# Epigenetic Alterations Related to Gestational Diabetes Mellitus

**DOI:** 10.3390/ijms22179462

**Published:** 2021-08-31

**Authors:** Jorge Valencia-Ortega, Renata Saucedo, Martha A. Sánchez-Rodríguez, José G. Cruz-Durán, Edgar G. Ramos Martínez

**Affiliations:** 1Unidad de Investigación Médica en Enfermedades Endocrinas, UMAE Hospital de Especialidades, Instituto Mexicano del Seguro Social, Mexico City 06600, Mexico; j.valencia.o@hotmail.com; 2Unidad de Investigación en Gerontología, Facultad de Estudios Superiores Zaragoza, Universidad Autónoma de México, Mexico City 04510, Mexico; masanrod@yahoo.com; 3UMAE Hospital de Gineco-Obstetricia No. 3, Instituto Mexicano del Seguro Social, Mexico City 06600, Mexico; 27jgcd@gmail.com; 4Universidad Autónoma Benito Juárez de Oaxaca and Instituto de Cómputo Aplicado en Ciencias, Oaxaca 68120, Mexico; edgargus2@gmail.com

**Keywords:** gestational diabetes, epigenetics, DNA methylation, miRNAs, adipose tissue, placenta

## Abstract

Gestational diabetes mellitus (GDM) is the most common metabolic complication in pregnancy, which affects the future health of both the mother and the newborn. Its pathophysiology involves nutritional, hormonal, immunological, genetic and epigenetic factors. Among the latter, it has been observed that alterations in DNA (deoxyribonucleic acid) methylation patterns and in the levels of certain micro RNAs, whether in placenta or adipose tissue, are related to well-known characteristics of the disease, such as hyperglycemia, insulin resistance, inflammation and excessive placental growth. Furthermore, epigenetic alterations of gestational diabetes mellitus are observable in maternal blood, although their pathophysiological roles are completely unknown. Despite this, it has not been possible to determine the causes of the epigenetic characteristics of GDM, highlighting the need for integral and longitudinal studies. Based on this, this article summarizes the most relevant and recent studies on epigenetic alterations in placenta, adipose tissue and maternal blood associated with GDM in order to provide the reader with a general overview of the subject and indicate future research topics.

## 1. Introduction

GDM is defined as any degree of glucose intolerance first identified during pregnancy [1]. The prevalence of this disorder varies globally, depending on the diagnostic criteria applied and on the ethnic group studied. It varies from 1% to more than 30% [2]. The population with GDM has been increasing worldwide, mainly as a result of the rising proportion of women with pre-pregnancy obesity, a sedentary lifestyle and advanced maternal age at birth. Additional risk factors include family history of diabetes, parity, previous GDM, prior macrosomia and excessive weight gain in pregnancy. GDM is associated with short-term and long-term adverse consequences in the mother and in the offspring. In the mother, it increases the risk of cesarean section and favors the development of gestational hypertension or pre-eclampsia. Furthermore, it has been estimated that women with GDM have a more than sevenfold increased risk of type 2 diabetes mellitus (T2DM) compared to normoglycemic pregnancies. In the fetus, the risk of macrosomia or a larger size for gestational age increases, which leads to complications such as shoulder dystocia and Erb’s palsy. Another important neonatal complication is immediate hypoglycemia at birth caused by fetal hyperinsulinemia [3,4]. In adulthood, the child is more likely to develop obesity, T2DM and cardiovascular disease [5].

In addition to lifestyle factors, genetic heritability is implicated in the etiology of GDM, and emerging data indicate contributions of environmental and dietary factors to the risk of developing GDM, through epigenetic changes. Epigenetics refers to changes in gene expression that occur without any alterations in the nucleotide sequence of DNA. Epigenetic changes include DNA methylation, chemical histone modifications (methylation, acetylation, etc.) and non-coding RNA molecules such as micro RNAs (miRNAs) and long non-coding RNAs [6,7]. The current evidence of epigenetic factors related to GDM is substantial and allows us to see that there are particular epigenetic signatures before the onset of hyperglycemia; during GDM and postpartum; and before the development of T2DM. Thus, the objectives of this review are to present an organized and updated summary of epigenetic alterations related to pathogenesis of GDM and indicate future topics for research.

## 2. The Pathophysiology of GDM

To understand the pathophysiology of GDM, it is necessary to know the metabolic changes that occur in healthy pregnancy. The first trimester of pregnancy is characterized by a progressive increase in insulin secretion, perhaps as a result of an increase in pancreatic β-cell mass, and increased sensitivity to this hormone. From the second trimester, a progressive decrease in insulin sensitivity, called insulin resistance, begins, mainly in the adipose tissue, skeletal muscle and liver [8,9]. Historically, this insulin resistance has been thought to be the result of increased concentrations of placental hormones such as human placental lactogen (hPL) and placental growth hormone, as their effects can interfere with insulin receptor signaling and cause a marked decrease in glucose utilization [10]. To compensate for this insulin resistance, the pancreatic β cells increase insulin secretion, thereby preventing the blood glucose concentration from exceeding normal values [11]. In addition to these hormonal changes, some studies have suggested the participation of adipose tissue in insulin resistance in pregnancy. During pregnancy, there is an increase in fat mass that can vary from 2 to 5 kg [12]. Adipose tissue functions as an endocrine organ, in which increase in fat is related to changes in the production of cytokines and adipokines, such as interleukin 1 beta (IL-1β), interleukin 6 (IL-6), tumoral necrosis factor alpha (TNF-α), leptin and adiponectin, whose effects intervene in insulin sensitivity in pregnancy [13].

The pathophysiology of GDM is still under investigation; however, it is characterized by two main mechanisms which result in hyperglycemia: dysfunction of pancreatic β cells and exacerbated insulin resistance. In women who develop GDM, there is a defect in the response of β cells that compensates for insulin resistance, which becomes evident in late pregnancy, when insulin resistance is greatest [14]. There is evidence that this defect in β cell function exists before pregnancy, but only becomes clinically apparent with the increased insulin resistance of pregnancy. For this reason, screening for GDM is usually done between weeks 24 and 28 of pregnancy, and there are various criteria for GDM diagnosis using the oral glucose tolerance test (OGTT) [15].

In addition, insulin resistance in women with GDM may be more pronounced due to the down-regulation of adiponectin and up-regulation of leptin and pro-inflammatory cytokines [16]. Interestingly, in many cases, insulin resistance already exists before pregnancy, especially in populations with high rates of obesity [17].

There is emerging evidence that epigenetic changes contribute to the pathophysiology of GDM. Recently, several studies have reported placental and adipose tissue epigenetic changes associated with GDM. In addition, GDM has been associated with epigenetic modifications that are detectable in maternal blood. A summary of the methodological characteristics of studies that compared epigenetic changes in placental tissue, adipose tissue and peripheral blood between GDM patients and controls is presented in Table 1.

## 3. Epigenetic Alterations Determined in the Placenta

The placenta secretes steroids, hormones and cytokines that are necessary for the correct course of pregnancy [40,41]. In GDM, the size and weight of the placenta are significantly greater than in normal pregnancy [42]. This increase in placental size is accompanied by higher concentrations of hPL, estradiol, and progesterone in the maternal circulation, which can promote the hyperglycemic state [43,44,45]. Moreover, GDM has a placental inflammatory component characterized by increased production of interleukin-8 (IL-8), TNF-α and leptin [46].

Interestingly, the diabetic placenta shows alterations in the proliferation, apoptosis and control of the trophoblast cell cycle [47]. Next, placental epigenetic profiles are described, some of which explain, in part, some of these associations between the placenta and GDM. The main findings are summarized in Figure 1.

### 3.1. DNA Methylation

DNA methylation occurs when a methyl group is covalently added to 5-carbon of the cytosine in DNA by enzymes called DNA methyltransferases. Generally, this modification occurs when cytosine is followed by guanine, which is known as a CpG dinucleotide (the “p” refers to the phosphate group). Approximately 70–80% of the CpGs in the human genome are methylated, and the majority of the remainder are in gene promoter regions in clusters of CpGs called CpG islands that can be hundreds to tens of thousands of base pairs in length [48]. Generally, the level of DNA methylation in the promoter region of the gene inversely correlates with gene transcription, since methylation provides steric hindrance and favors compaction of chromatin, physically preventing DNA-binding proteins from recognizing their target sequences. CpG dinucleotides and CpG islands can also be located within genes and influence transcriptional activity; for example, when CpGs are close to or in splice sequences, methylation can influence the splice site and lead to alternative isoforms of mature mRNA [49,50,51]. Methylation can also occur in enhancer elements, which promotes the binding of proteins such as methyl-binding domain proteins, favoring transcription [52,53], or in repressor elements, which prevents the binding of methylation-sensitive proteins and leads to improved transcription, a rare case in which methylation positively correlates with transcription [54,55].

Lesseur et al. [18] evaluated the methylation levels of *leptin* (*LEP*) promoter in placental tissues from women with GDM in comparison to non-GDM women. The results showed that the average methylation of 23 CpGs sites analyzed in the *LEP* promoter was 2.5% higher in women with GDM. Interestingly, maternal pre-pregnancy obesity was not a significant predictor of *LEP* methylation, but it was strongly associated with GDM. Thus, the authors concluded that GDM mediates pre-pregnancy obesity’s effects on placental *LEP* methylation. In contrast, in the study by Gagné-Ouellet et al. [19] it was observed that higher maternal glycemia during pregnancy decreased methylation levels of *LEP* in the GDM placenta. The authors identified one CpG site (cg15758240) with more convincing results about the impact of maternal glucose levels on placental *LEP* DNA methylation. This CpG site has been characterized as an *LEP* interaction region located upstream of the *LEP* gene, with important roles for transcriptional regulation. In these studies, the placental expression of leptin was not measured, so the possible effect of differential methylation on mRNA and umbilical cord levels remains uncertain.

Bouchart et al. [20] analyzed the placental DNA methylation levels of *adiponectin* (*ADIPOQ*) in pregnant women classified according to glucose tolerance status in impaired glucose tolerance women (IGT) and normal glucose tolerant women (NGT). Although the authors did not observe significant differences in DNA methylation levels between the groups, they found that lower DNA methylation levels in the promoter of *ADIPOQ* on the fetal side of the placenta were correlated with higher maternal glucose levels during the second trimester of pregnancy. Furthermore, lower DNA methylation levels on the maternal side of the placenta were associated with higher insulin resistance during the second and third trimesters of pregnancy. Finally, on the maternal side of placenta, higher DNA methylation levels were associated with lower maternal circulating adiponectin levels throughout pregnancy and postdelivery.

A recent body of evidence suggests that serotonin, a multifunctional signaling molecule, contributes to obesity and related metabolic disorders, although the mechanisms are not well understood. The serotonin transporter, an integral membrane protein, mediates uptake of serotonin into cells, regulating the serotonin homeostasis [56,57]. Blazevic et al. [21] analyzed the methylation and mRNA levels of the placental serotonin transporter gene, also known as the *solute carrier family 6 member 4* (*SLC6A4*), in placentas from GDM and NGT women. They observed that the DNA methylation across the seven analyzed loci was decreased in the GDM group in comparison to the control group and inversely correlated with placental *SLC6A6* mRNA levels. Interestingly, maternal plasma glucose levels in the 24th to 28th weeks of gestation were negatively correlated with average DNA methylation.

Brown adipose tissue (BAT) is suspected to protect against obesity. Côté et al. [22] assessed DNA methylation variations in genes involved in BAT genesis and activation. They evaluated the DNA methylation levels at *PR domain-containing protein 16* (*PRDM16*), *bone morphogenetic protein 7* (*BMP7), C-terminal binding protein 2 (CTBP2)* and *peroxisome proliferator-activated receptor-gamma co-activator 1 alpha (PPARGC1α*) gene loci in the fetal side of placentas from 133 women (33 with GDM) followed during pregnancy (one visit at the end of each trimester), and observed that only *BMP7* DNA methylation levels were lower in GDM compared to NGT. Interestingly, when analyzing all the women, higher *PPARGC1α* DNA methylation levels were associated with higher second trimester fasting glucose levels and 2 h post-OGTT glycemia; lower *PRDM16* DNA methylation levels were associated with higher fasting glucose levels in the second and third trimesters, and lower *BMP7* DNA methylation levels were associated with higher 2 h post-OGTT glycemia. In addition, Wang et al. [23] analyzed the methylation status and expression levels of the *PPARGC1α* and the *pancreatic and duodenal homeobox l* (*PDX1*) in placental tissue. They observed that the methylation frequency of *PPARGC1α* was higher and its expression was lower in GDM group compared to controls. There were no significant differences in relation to *PDX1*. Interestingly, *PPARGC1α* is a transcriptional co-activator that participates in lipid and carbohydrate metabolism in many tissues, including adipose tissue, muscle and liver.

Houde et al. [24] analyzed the placental methylation levels of the *lipoprotein lipase* (*LPL*) enzyme, which contributes to the transfer of free fatty acids from maternal lipoproteins to the fetus, and observed that the women with GDM had lower levels of methylation in each of the 3 CpG sites evaluated in comparison to NTG women. In addition, methylation levels of CpG2 and CpG3 sites (both located within the intronic CpG island) were negatively correlated with placental *LPL* mRNA levels, which were higher in GDM compared to NGT women. Interestingly, after adjusting for all potential confounding variables, it was observed that methylation levels at CpG1 (located within the promoter region) were negatively correlated with 2-h post-OGTT glucose concentrations, and CpG3 methylation levels negatively correlated with maternal concentrations of HDL-cholesterol during the third trimester of pregnancy.

### 3.2. miRNAs

miRNAs are small non-coding RNA sequences about 22 nucleotides in length that are capable of regulating gene expression. Most miRNAs are transcribed from DNA into primary sequences that are processed to give rise to mature miRNAs [58]. In most cases, miRNAs are loaded onto the Argonaute protein to form the minimal miRNA-induced silencing complex (miRISC). By base complementarity, miRNA leads to the interaction of miRISC with the 3′ untranslated region (UTR) of its target mRNA. If the complementarity is complete, the miRNA–mRNA interaction induces the endonuclease activity of Argonaute and the mRNA is degraded. In the case that the complementarity is lower, the miR–SC complex only inhibits the translation of the mRNA [59].

In the study of Qiu et al. [25] up-regulated expression of *miR-518d* in placenta was observed from women with GDM in comparison to the normal pregnancy group. Moreover, the authors identified increased mRNA levels of *nuclear factor-kappa B* (*NF-κB*), *cytochrome C oxidase subunit II* (*COX-2*), *TNF-α, IL-1β* and *IL-6*; and decreased mRNA levels of *peroxisome proliferator-activated receptor α* (*PPARα*) in placentas, in women with GDM. Using cultures of the human placental trophoblast cell line HTR8/SVneo transfected with *miR-518d* mimetics or inhibitors, the authors demonstrated that *miR-518d* promotes the mRNA expression of *COX-2, TNF-α, IL-1β* and *IL-6*. Furthermore, through luciferase assays, *PPARα* was validated as the target gene of *miR-518d*, and the decreased mRNA and protein levels of *PPARα* and its downstream genes in cultures transfected with *miR-518d* mimetics, suggesting that *PPARα* is negatively regulated by *miR-518d*. Inhibition of *PPARα* mRNA using *PPARα*-specific antagonist and *PPARα* siRNA interference resulted in increased mRNA and protein expression of *NF-κB, COX-2, TNF-α, IL-1β* and *IL-6*. Another series of experiments showed that the level of *NF-κB* in the nucleus was elevated with a high glucose concentration. It was reduced with the application of *miR-518d* inhibitors, and was increased with the knockdown of *PPARα*. This suggests that the development of GDM might be associated with an inflammatory response in the placenta which is regulated by *miR-518d* through the inhibition of *PPARα* and the activation of *NF-κB.*

Sun et al. [26] observed that the placental expression levels of *miR-29b* were lower in a GDM group compared to controls. Moreover, by using the trophoblast cell line HTR8/SVneo with an *miR-29b* mimetic or inhibitor, the authors observed that *miR-29b* over-expression inhibited cell growth, late apoptosis, migration and invasion; and *miR-29b* knockdown promoted cell migration and invasion. This suggests that, in women with GDM, the proliferation and infiltration abilities of trophoblast cells are strengthened. Using luciferase assays, the authors demonstrated that the binding site in the three prime untranslated region (3′-UTR) of *hypoxia inducible factor 3A* (*HIF3A*) is specific for *miR-29b*, and that *miR-29b* inversely regulates *HIF3A* expression in the trophoblast cell. In summary, *miR-29b* regulates trophoblast cells activities by modulating *HIF3A* expression.

Cao et al. [27] showed that *miR-98* levels were up-regulated in placentas from women with GDM compared with normal placentas. By using JEG-3 cells transfected by *miR-98* mimetics or inhibitors and measuring the global DNA methylation level, it was observed that *miR-98* can positively regulate the global DNA methylation level. Further investigations revealed that *methyl CpG binding protein 2* (*MECP2*) was the target gene of *miR-98*. In JEG-3 cells, the level of MECP2 protein was significantly down-regulated by an *miR-98* mimetic and up-regulated by an *miR-98* inhibitor, suggesting that *MECP2* is negatively regulated by *miR-98*. These findings were supported by decreased expression of *MCEP2* in placentas from women with GDM. To investigate the relationships of *miR-98* and *MCEP2* with the global DNA methylation level, the effects of the former two on the expression of DNA methyltransferase was evaluated. *miR-98* mimetic and *MECP2* siRNA increased the protein levels of DNA methyltransferase 1 (DNMT1), and *MECP2* expression vector decreased DNMT1 protein level. Further investigations showed that *miR-98* mimetic and *MECP2* siRNA reduced the mRNA and protein levels of *transient receptor potential 3* (*TRPC3*), one of the target genes of *MECP2* involved in vasoconstriction and the regulation of blood pressure in metabolic syndrome [60], and the *miR-98* inhibitor and *MECP2* expression vector increased *TRPC3* expression. These findings were supported by decreased mRNA and protein levels of *TRPC3* in placentas from women with GDM, but only in certain age ranges. Thus, the authors demonstrated that *miR-98* is involved in the occurrence of GDM through the *MECP2*-*TRPC3* pathway.

Ding et al. [28] conducted an integrative analysis of the gene expression profiles of both mRNAs and miRNAs in placental tissues from eight women with GDM and eight controls: 281 mRNAs and 32 miRNAs with differential expression were observed. Among them, eight mRNAs (*TBL1X, NOTUM, FRMD4A, SLC16A2, CLDN19, CCL18, HTRA1* and *SLC39A6*) and five miRNAs (*miR-202-5p, miR-138-5p, miR-210-5p, miR-3158-5p and miR-4732-3p*) were validated by quantitative reverse transcription polymerase chain reaction (qRT-PCR) in the entire study sample. The functional analysis based on the results of the two massive analyses identified a significant up-regulation of two biological functions: cellular development and cellular movement. Furthermore, the paired mRNA–miRNA analysis with 91 negatively expressed mRNA targets for 13 miRNAs allowed the construction of molecular interactions networks. The most enriched networks were related to cellular development and functions, organ morphology and organismal development. Notably, *miR-138-5p* was a central node in this network. The overexpression of *miR-138-5p* reduced both the migratory ability and the proliferation of HTR-8 cells, whereas the inhibition of *miR-138-5p* increased both the migratory ability and the proliferation of HTR-8 cells. Using luciferase assays, it was shown that *miR-138-5p* targets *TBL1X* and suppresses *TBL1X* expression in HTR-8 cells. Further investigations with HTR-8 cells co-transfected with *miR-138-5p* inhibitor and *TBL1X* siRNA indicated that *miR-138-5p* could inhibit the migration and proliferation of HTR-8 trophoblasts by targeting *TBL1X*. Both mRNA and protein levels of *TBL1X* were significantly up-regulated in GDM placentas and a significant negative correlation between the expression of *miR-138-5p* and *TBL1X* mRNA was observed. Interestingly, the placentas in women with GDM were heavier than in controls. The weight was negatively correlated with the expression of *miR-138-5p* and positively correlated with the mRNA levels of *TBL1X*. Thus, reduced expression of *miR-138-5p* contributed to the excessive growth of the placenta in GDM by enhancing the proliferation of trophoblasts by targeting *TBL1X*.

### 3.3. Histone Modifications

Genomic DNA is condensed into chromatin, which consists of said DNA bound to histone proteins. There are four types of histones that form an octamer in which DNA is wrapped, all of which constitutes a nucleosome. Histone tails can have chemical modifications, including acetylation, methylation and phosphorylation, among others, depending on the activity of different enzymes that add or remove such modifications [61,62]. These modifications alter the interactions of DNA with histones, which in turn can increase or decrease access to genes, thereby affecting gene expression. The most studied chemical modifications are acetylation and methylation (which can occur in the form of mono, di and tri-methylation), which occur mainly on lysine residues [63].

Hepp et al. [29] analyzed the levels of lysine acetylation of histones H3 at the H3K9 site (lysine at position 9 of histone H3) and the levels of lysine tri-methylation at the H3K4 site in nuclei of villous syncytiothrophoblast cells (SCT) and extra villous trophoblast cells (EVT) from term placentas from GDM women and controls. In both SCT and EVT cells, the levels of lysine acetylation at the H3K9 site were lower in GDM women in comparison to controls. The levels of lysine tri-methylation at H3K4 site were similar between the groups. Since an association between vitamin D deficiency and GDM has been documented [64], the authors, using cell cultures, evaluated the effects of different concentrations of vitamin D on acetylation levels of lysine at the H3K9 site, and observed that acetylation levels were not affected by low doses of calcitriol and decreased slightly at the highest concentration.

## 4. Epigenetic Alterations Determined in Adipose Tissue

The main epigenetic alterations in adipose tissue are summarized in Figure 2.

### 4.1. DNA Methylation

Rancourt et al. [30] evaluated the mRNA expression of *TNF-α* and *suppressor of cytokine signaling 3* (*SOCS3*), two important components in the inflammation and apoptosis processes in adipose tissue [65,66], and the methylation in their promoters in subcutaneous and visceral adipose tissues from women with GDM in comparison with women with NGT. It was observed that the mRNA expression levels of *TNF-α* and *SOCS3* were significantly higher only in visceral adipose tissue from women with GDM compared to controls. Curiously, circulating maternal *TNF-α* levels correlated with both visceral adipose tissue *TNF-α* and *SOC3* mRNA levels. In visceral adipose tissue, the methylation analysis showed decreased methylation percentages for three CpGs sites of the *TNF-α* promoter in the GDM group, and the methylation on the *SOCS3* promoter was not different between the groups. No significant correlation was observed between methylation and mRNA levels of *TNF-α*. Assuming that *TNF-α* mRNA levels are not regulated by promotor methylation, there are other mechanisms that can regulate this transcript, such as the repressive effect of *miR-19a-3p* or the transcript stabilization by Hu antigen R [67].

Hypoadiponectinemia has been observed in people with insulin resistance, T2DM, GDM and obesity [68,69]. Adipose tissue is the main source of adiponectin, and recently, it has been suggested that the production of this hormone is greater in subcutaneous adipose tissue than in visceral adipose tissue [70]. This motivated Ott et al. [31] to evaluate adiponectin plasma concentrations and mRNA and DNA methylation levels in blood cells, subcutaneous adipose tissue and visceral adipose tissue from women with GDM compared to controls. Women with GDM showed hypoadiponectinemia accompanied by decreased mRNA levels in both adipose tissue types. There were greater differences between visceral adipose tissues than subcutaneous adipose tissues. Maternal adiponectin levels positively correlated with mRNA levels from both visceral and subcutaneous adipose tissues, but the correlation was stronger with subcutaneous adipose tissue mRNA levels. Four CpGs sites were at the *ADIPOQ* locus had different methylation levels (two in visceral adipose tissue and two in maternal blood) in the groups, but only the R3 CpG1 site in visceral adipose tissue, which showed greater methylation in GDM, inversely correlated with mRNA levels of *adiponectin.* This study highlights the importance of the two types of adipose tissue in the pathophysiology of GDM.

Deng et al. [32] while analyzing the global DNA methylation and whole genome expression in visceral omental adipose tissue from women with GDM and controls, observed 485 down-regulated and 485 up-regulated genes, and 1298 hypomethylated genes and 1568 hypermethylated genes in the GDM group. After integrating the data from the two massive techniques and validating the results, only three genes (*HLA-DMB, MSLN* and *HSPA6*) showed significantly different methylation and expression levels, and only *MSLN* showed a strong negative correlation between expression and methylation levels. Based on the integration of Gene Ontology (GO), Kyoto Encyclopedia of Genes and Genomes (KEGG) and Genomes database pathway analysis of expression and methylation profiles, the authors found that the antigen processing and presentation pathway and immune-related genes were closely associated with GDM in visceral adipose tissue.

### 4.2. MiRNAs

Shi et al. [33] while analyzing the global expression of miRNAs in omental adipose tissue in women with GDM against controls, observed differential expression of 17 miRNAs. With qRT-PCR, only the increased expression of *miR-222* in GDM was validated. Interestingly, the authors observed that the expression of *miR-222* negatively correlated with the protein expression of estrogen receptor alpha (ERα) and glucose transporter 4 (GLUT4) in this same tissue. When observing that estradiol levels were higher in the GDM group, the authors used 3T3-L1 murine adipocyte cultures and observed that the increase in the concentration of 17β-estradiol in the medium increased the gene expression of *miR-222* and decreased the protein expression of ERα and GLUT4. Through silencing assays in this same line cell, the authors observed that the silencing of *miR-222* resulted in increased protein expression of ERα (later, by luciferase assays, it was shown that *miR-222* was capable of binding to the 3′-UTR of *ERα* mRNA) and augmented glucose consumption after insulin stimulation—about 40% higher than in non-silenced cells. This was due to higher synthesis and translocation of GLUT-4. Thus, the authors propose that *miR-222* plays an important role in insulin resistance through the regulation of ERα and GLUT-4.

To our knowledge, there are no studies on histone chemical modifications in adipose tissue associated with GDM.

## 5. Epigenetic Alterations Determined in Blood

It is important to point out that in most studies that have analyzed epigenetic profiles in blood, whether by analyzing DNA, histones or miRNAs, a relationship between the findings and the pathophysiology of GDM could not be established. Therefore, the studies are limited to describing epigenetic characteristics associated with GDM.

### 5.1. DNA Methylation

Dias et al. [34] evaluated the peripheral blood DNA methylation patterns in women with GDM and in NGT sampled around week 19 of gestation, and identified 1046 differentially methylated CpG sites (148 were hypermethylated and 898 were hypomethylated) in those women who developed GDM. The authors selected the top five significantly differentially methylated CpG sites for further analysis. These five CpG sites were associated with four unique genes, including *solute carrier family 9 member A3* (*SLC9A3*), *male-enhanced antigen 1 and kelch domain-containing protein 3* (*MEA1* and *KLHDC3*), *calmodulin binding transcription activator 1* (*CAMTA1*), *RAS P21 protein activator 3* (*RASA3*) and one unknown gene. The KEGG analysis identified canonical pathways related to signal transduction, cell growth, proliferation, differentiation and apoptosis, insulin resistance, glucose metabolism, inflammation, neurological signaling and oncogenesis. The GO analysis identified biological processes associated with structural organization and development, and molecular functions associated with regulatory or binding activities. In this study, pyrosequencing validation was not performed, so these potential epigenetic markers need to be validated longitudinally in a larger population.

Wu et al. [35] in gestational weeks 12–16, analyzed the global maternal blood DNA methylation patterns in a cohort of 22 women (11 who developed GDM, and their respective matched controls) and identified 100 CpG sites (comprising 66 genes) differentially methylated between women who developed GDM and healthy pregnant women. After using rigorous criteria for the absolute β-value differences across all 11 matched pairs and validating by bisulfite pyrosequencing (BSP), the *constitutive photomorphogenic homolog subunit 8* (*COPS8*), *phosphoinositide-3-kinase, regulatory subunit 5* (*PIK3R5*), *3-hydroxyanthranilate 3,4-dioxygenase* (*HAAO*), *chromosome 5 open reading frame 34* (*C5orf34*) and *coiled-coil domain containing 124* (*CCDC124*) genes, each with varied functions, were the only ones that maintained differential methylation between the groups. Thus, the authors proposed that changes in the methylation of these genes before the diagnosis of GDM have potential use as predictive biomarkers of this disorder, although they did not perform any type of analysis in this regard.

### 5.2. Histone Chemical Modifications

Michalczyk et al. [36] analyzed the levels of lysine di-methylation of histones H3 at sites H3K27 (lysine at position 27 of histone H3), H3K4, H3K79, H3K36 and H3K9 in DNA from white blood cells from pregnant women evaluated at week 30 of gestation and at weeks 8–10 and 20 postpartum. The women were classified as non-diabetic, GDM who did not develop T2DM, GDM who developed T2DM and women with pre-existing T2DM. Just focusing on the results on GDM, the authors observed that di-methylation in H3K4 was approximately 80% lower in women with GDM who developed T2DM compared to women with GDM who did not develop T2DM by week 10 (roughly) postpartum, and that di-methylation in H3K27 was approximately 50% lower in women with GDM who developed T2DM compared to women with GDM who did not develop T2DM by week 20 postpartum. Thus, the authors propose that these changes in histone methylation have potential utility as predictive biomarkers for the development of T2DM in women with GDM, but they did not carry out analyses to support this proposal.

### 5.3. MiRNAs

Zhao et al. [37] analyzed miRNAs in maternal serum from women who developed GDM and their respective controls. Both groups were sampled in one gestational week from 16 to 19. They identified significantly lower expression of *miR-132*, *miR-29a* and *miR-222* in women who developed GDM, but only *miR-29a* and *miR-222* were externally validated. Further investigations in HepG2 cells showed that *insulin-induced gene 1* (*INSIG1*) was the target gene of *miR-29a* and that the knockdown of *miR-29a* increases the protein levels of *INSIG1*. Since *INSIG1* has become related to sterol regulatory element-binding protein (SREBP)-mediated regulation of *phosphoenolpyruvate carboxy kinase 2* (*PCK2*), a key enzyme in hepatic gluconeogenesis, the authors evaluated the expression of *PCK2* in this same knockdown model and observed increased mRNA levels of *PCK2*. This suggests that *miR-29a* is a negative regulator of serum glucose. A recent study supports the observation that GDM is characterized by lower circulating levels of *miR-29a* [71].

Zhu et al. [38] constructed two small circulating RNA libraries from pooled plasma, one from women who subsequently developed GDM and another from controls, all sampled in gestational a week from 16 to 19. The authors identified 32 miRNAs with differential expression; however, only five miRNAs (*hsa-miR-16-5p, hsa-miR-17-5p, hsa-miR-19a-3p, hsa-miR-19b-3p* and *hsa-miR-20a-5p*) with increased expression levels in GDM were validated by qRT-PCR. The target prediction, GO analysis and pathway identification of these five miRNAs allowed the construction of a regulatory network with seven target genes (*MAPK-1, IRS-1, IRS-2, SOS-1, SMAD5, SMAD4* and *AKT3*) and five signaling pathways (mitogen-activated protein kinase, insulin, type 2 diabetes mellitus, transforming growth factor β and mammalian target of rapamycin) for GDM. The results of Zhu et al. [38] regarding *hsa-miR-20a-5p* differ from those of Pheiffer et al. [39] who observed lower circulating levels of *hsa-miR-20a-5p* in women with GDM compared with controls; moreover, this miRNA and one or more risk factor were significant predictors of GDM.

Qiu et al. [25] while analyzing the expression levels of *miR-518d* in peripheral plasma and placentas from women with normal pregnancies and women with GDM, sampled between the 36th and 42th week of pregnancy, observed up-regulated expression of *miR-518d* in plasma from women with GDM in comparison to controls.

## 6. Conclusions

Several studies have investigated epigenetic changes in the placentas, adipose tissue and peripheral blood of GDM patients. Some authors have found alterations in the methylation of genes that modulate insulin resistance during pregnancy. Others have found alterations in miRNAs involved in inflammation, insulin resistance and metabolism. Notably, other epigenetic mechanisms, such as chemical histone modification, have been reported as predictive biomarkers for the development of T2DM at postpartum. These findings suggest that epigenetic modifications are related to the etiology of GDM and to its progression to T2DM. However, causality was not established in the majority of the studies, and it is therefore not clear whether epigenetic changes predate GDM. Interestingly, some epigenetic changes correlate with glucose levels and may be indicators of metabolic dysfunction. On the other hand, most studies did not explore whether epigenetic modifications change gene expression, and did not validate their findings. Therefore, additional studies seeking causality are necessary to elucidate the underlying mechanisms for the epigenetic changes and to find out how epigenetic changes regulate pathways crucial for the pathophysiology of GDM.

## Figures and Tables

**Figure 1 ijms-22-09462-f001:**
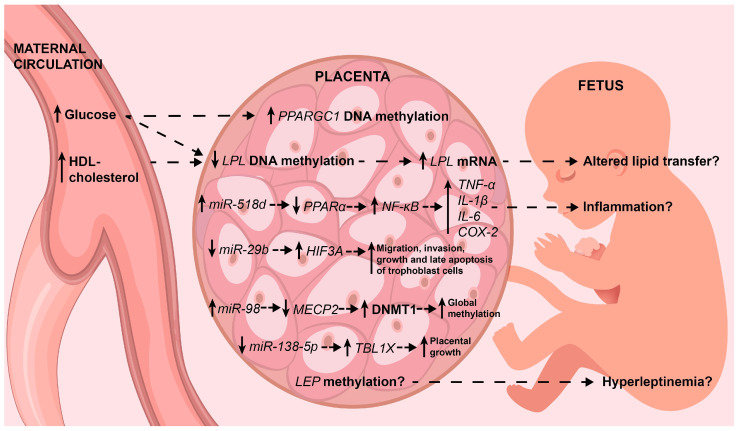
The main placental epigenetic alterations associated with GDM. Epigenetic alterations and their effects on target genes are shown (see the text for deeper details). Interestingly, some explain, in part, the well-known pro-inflammatory status and excessive growth of the placenta. On tissue, solid arrows indicate up-regulation or down-regulation in maternal circulation and the fetus indicate low or high circulating levels. Dotted line arrows indicate association. HDL-cholesterol: high-density lipoprotein cholesterol; mRNA: messenger ribonucleic acid; FFA: free fatty acids; *PPARα*: proliferator-activated receptor alpha; *NF-κB*: nuclear factor-kappa B; *TNF-α*: tumoral necrosis factor alpha; *IL-1β*: interleukin 1 beta; *IL-6*: interleukin 6; *COX-2*: cytochrome C oxidase subunit II; *HIF3A*: hypoxia inducible factors 3A; *MECP2*: methyl CpG binding protein 2; DNMT1, DNA methyltransferase 1; *TBL1X*: transducin β–like protein 1; *LEP*: leptin.

**Figure 2 ijms-22-09462-f002:**
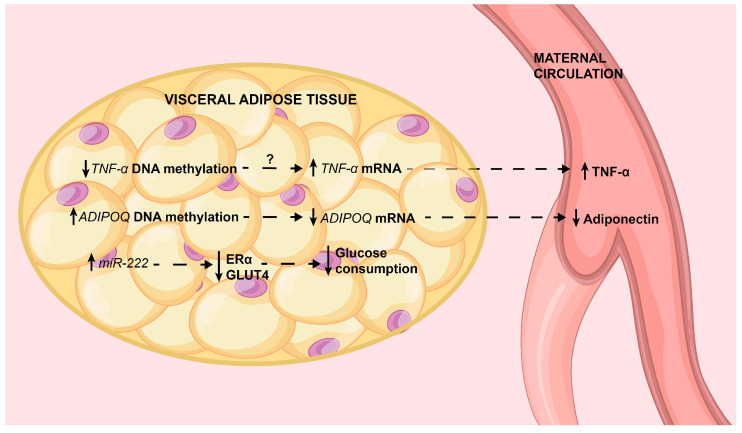
The main epigenetic alterations in visceral adipose tissue associated with GDM. Epigenetic alterations and their effects on target genes are shown (see the text for deeper details). These findings explain in part the well-known pro-inflammatory status, decreased production of adiponectin and insulin resistance in visceral adipose tissue. In tissue, solid line arrows indicate or up-regulation or down-regulation; in maternal circulation they indicate low or high circulating levels. Dotted arrows indicate associations. *TNF-α*: tumoral necrosis factor alpha; *ADIPOQ*: adiponectin; ERα: estrogen receptor alpha; GLUT4: glucose transporter 4.

**Table 1 ijms-22-09462-t001:** The methodological characteristics of studies that compared epigenetics changes between GDM patients and controls.

Authors	Study Design	n	Method Used for Epigenetic Analysis	Biological Sample Used for Epigenetic Analysis	Genes or Proteins Affected	GDM Criteria
Lesseur, et al. [18]	Cohort	47 GDM and 488 non GDM women	BSP	Placental tissue (fetal side)	*LEP*	Not reported
Gagné-Ouellet, et al. [19]	Cohort	12 GDM and 250 non GDM women	Infinium MethylationEPIC BeadChip	Placental tissue (fetal side)	*LEP*	Canadian Diabetes Association (year not reported)
Bouchart, et al. [20]	Cohort	31 IGT women and 67 NGT women	BSP	Placental tissue from both maternal and fetal sides	*ADIPOQ*	Only 2 women from IGT group fulfilled the ADA 2009 criteria for GDM. Women were classified as having IGT with a 2-h post-OGTT glucose level ≥7.8 mmol/L
Blazevic, et al. [21]	Case-control	18 GDM and 32 NGT	DBS	Placental tissue (fetal side)	*SLC6A4*	IADPSG 2010
Côté, et al. [22]	Cohort	33 GDM and 100 NGT	BSP	Placental tissue (fetal side)	*PRDM16, BMP7, CTBP2* and *PPARGC1α*	WHO 2013
Wang, et al. [23]	Case-control	18 GDM and 32 control	DBS	Placental tissue (fetal side)	*PPARGC1α* and *PDX1*	Two step: Step 1: 50 g GCT Cutoff value (mmol/L): ≥7.8 Step 2: 100 g OGTT Cutoff value (mmol/L) Fasting: >5.6 1 h: >10.3 2 h: >8.6 3 h: >6.7 Diagnosed: 4 abnormalities
Houde, et al. [24]	Cohort	27 GDM and 99 NGT	Illumina Human Methylation 450 DNA Analysis Beadchip and BSP	Placental tissue (fetal side)	*LPL*	WHO 2013
Qiu, et al. [25]	Case-control	60 GDM and 60 controls	qRT-PCR	Plasma from maternal peripheral blood and placenta	*miR-518d*, *NF-κB, COX-2, TNF-α, IL-1β, IL-6* and *PPARα*	Not reported
Sun, et al. [26]	Case-control	204 GDM and 202 control	qRT-PCR	Placental tissue	*miR-29b* and *HIF3A*	WHO 2013
Cao, et al. [27]	Case-control	193 GDM and 202 control	qRT-PCR	Placental tissue	*miR-98, MECP2* and *TRPC3*	Not reported
Ding, et al. [28]	Case-control	Discovery stage: 8 GDM and 8 controls Validation stage: 28 GDM and 26 control	RNA-seq and qRT-PCR	Placental tissue	*miR-202-5p, miR-138-5p, miR-210-5p, miR-3158-5p, miR-4732-3p, TBL1X, NOTUM, FRMD4A, SLC16A2, CLDN19, CCL18, HTRA1* and *SLC39A6*	IADPSG 2010
Hepp, et al. [29]	Case-control	40 GDM and 40 controls	Immunohistochemistry and double immunofluorescence	Placental tissue (maternal side)	Histones H3	GSDB 2011
Rancourt, et al. [30]	Nested case-control	19 GDM and 22 controls matched for maternal age, socio-economic status, ethnic origin, parity and pre-pregnancy BMI	BSP	Subcutaneous and visceral adipose tissues	*TNF-α* and *SOCS3*	GSGO 2018
Ott, et al. [31]	Nested case-control	25 GDM and 30 controls matched for maternal age, ethnic origin, socio-economic status, parity and pre-pregnancy BMI	BSP	DNA from blood cells, subcutaneous and visceral adipose tissue	*ADIPOQ*	GSGO 2018
Deng, et al. [32]	Case-control	Discovery stage: 3 GDM and 3 controls Validation stage: 26 GDM and 24 controls	Illumina Human Methylation 450 k DNA Analysis Beadchip and BSP	Visceral omental adipose tissue	*HLA-DMB, MSLN*, and *HSPA6*	WHO 2013
Shi, et al. [33]	Case-control	Discovery stage: 3 GDM and 3 controls. Validation stage: 13 GDM and 13 controls.	AFFX miRNA expression chips and qRT-PCR	Omental adipose tissue	*miR-222*, ERα and GLUT4	ADA 2006
Dias, et al. [34]	Case-control	12 GDM and 12 controls matched for age, gestational age and BMI.	Illumina’s Infinium HumanMethylationEPIC Bead Chip	DNA from maternal peripheral blood	*SLC9A3, MEA1;KLHDC3, CAMTA1* and *RASA3*	IADPSG 2010
Wu, et al. [35]	Cohort	11 GDM and 11 controls matched for age, BMI, ethnicity, smoking, treatment, and folate supplementation	Illumina HumanMethylation450 BeadChip and BSP	DNA from maternal peripheral blood	*COPS8, PIK3R5, HAAO, C5orf34* and *CCDC124*	Not reported
Michalczy, et al. [36]	Cohort	27 pregnant women classified as non-diabetic (7 women), GDM who did not develop T2DM (8 women), GDM who developed T2DM (6 women), and women with pre-existing T2DM (6 women)	Western blot	White blood cells from maternal peripheral blood	Histones H3	IADPSG 2010
Zhao, et al. [37]	Nested case-control	Discovery stage: 24 GDM and 24 controls. Internal validation: 36 GDM and 36 controls. Two different external validations: 32 GDM and 32 controls in total. In all stages, the groups were matched for age, BMI, gestational age, and gravidity	TaqMan low density arrays Chips and qRT-PCR	Serum from maternal peripheral blood	*miR-29a, miR-222*, INSIG1 and PCK2	ADA 2004
Zhu, et al. [38]	Case-control	10 GDM and 10 controls matched for maternal age and gestational age	Ion Torrent high-throughput sequencing technology and qRT-PCR	Plasma from maternal peripheral blood	*hsa-miR-16-5p, hsa-miR-17-5p, hsa-miR-19a-3p, hsa-miR-19b-3p, hsa-miR-20a-5p*, *MAPK-1, IRS-1, IRS-2, SOS-1, SMAD5, SMAD4* and *AKT3*	ADA 2011
Pheiffer, et al. [39]	Case-control	28 GDM and 53 controls matched for age and BMI	qRT-PCR	Serum from maternal peripheral blood	*hsa-miR-20a-5p*	IADPSG 2010

BMI, body mass index; NGT, normal glucose tolerance; BSP, bisulfite pyrosequencing; DBS, direct bisulfite sequencing; qRT-PCR, quantitative reverse transcription—polymerase chain reaction; *LEP*, leptin; *ADIPOQ*, adiponectin; *SLC6A4*, solute carrier family 6 member 4; *PRDM16*, PR domain-containing protein 16; *BMP7*, bone morphogenetic protein 7; *CTBP2*, C-terminal binding protein 2; *PPARGC1α*, peroxisome proliferator-activated receptor-gamma, co-activator 1, alpha; *PDX1*, pancreatic and duodenal homeobox l; *LPL*, lipoprotein lipase; *NF-κB*, nuclear factor-kappa B; *COX-2*, cytochrome C oxidase subunit II; *TNF-α*, tumoral necrosis factor alpha; *IL-1β*, interleukin 1 beta; *IL-6*, interleukin 6; *PPARα*, peroxisome proliferator-activated receptor alpha; HIF3A, hypoxia inducible factors 3A; MECP2, methyl CpG binding protein 2; TRPC3, transient receptor potential 3; *TBL1X*, transducing beta like 1 X-linked; *NOTUM*, notum, palmitoleoyl-protein carboxylesterase; *FRMD4A*, FERM domain containing 4A; *SLC39A6*, solute carrier family 39 member 6; *SLC16A2*, solute carrier family 16 member 2; *CLDN19*, claudin 19; *CCL18*, C-C motif chemokine ligand 18; HTRA1, HtrA serine peptidase 1; *SOCS3*, suppressor of cytokine signaling 3; *HLA-DMB*, major histocompatibility complex, class II, DM beta; *MSLN*, mesothelin; *HSPA6*, heat shock protein family A (Hsp70) member 6; ERα, estrogen receptor alpha; GLUT4, glucose transporter 4; *SLC9A3*, solute carrier family 9 member A3; *MEA1;KLHDC3*, male-enhanced antigen 1; kelch domain-containing protein 3; *CAMTA1*, calmodulin binding transcription activator 1; *RASA3*, RAS P21 protein activator 3; *COPS8*, constitutive photomorphogenic homolog subunit 8; *PIK3R5*, phosphoinositide-3-kinase, regulatory subunit 5; *HAAO*, 3-hydroxyanthranilate 3,4-dioxygenase; *C5orf34*, chromosome 5 open reading frame 34; *CCDC124*, coiled-coil domain containing 124; INSIG1, insulin-induced gene 1; PCK2, phosphoenolpyruvate carboxy kinase 2; *MAPK-1*, mitogen-activated protein kinase 1; *IRS-1*, insulin receptor substrate 1; *IRS-2*, insulin receptor substrate 2; *SOS-1*, SOS Ras/Rac guanine nucleotide exchange factor 1; *SMAD5*, SMAD family member 5; *SMAD4*, SMAD family member 4; *AKT3*, AKT serine/threonine kinase 3; GSDB, German Society for Diabetes Mellitus; GSGO 2018, German Society for Gynecology and Obstetrics 2018; GCT, glucose challenge test; ADA, American Diabetes Association; WHO, World Health Organization; IADPSG, International Association of Diabetes and Pregnancy Study Group; IGT, impaired glucose tolerance.

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
