# Peer review of "Epigenetic Alterations Related to Gestational Diabetes Mellitus"

_ijms, 2021, doi:10.3390/ijms22179462_

Round 1

Reviewer 1 Report

In general, the manuscript is well written and organized.

I have one following comment, the authors present data on epigenetic changes in blood, placenta, and adipose tissue in GDM. In section 5. Epigenetic alterations determined in blood there are DNA methylation, miRNAs, and additionally, histone modifications described. Is there no published data on histone modifications in the adipose tissue and placenta available? If there are such studies maybe it will be worth mentioning it.

Author Response

The authors appreciate the valuable comments of reviewer. We performed a search on histone modifications in the placenta of women with GDM, and we found only one study on this alteration, which has been included in the text. We did not find data on this epigenetic modification in adipose tissue, so we have added a paragraph in section 4 that indicates the need for research on this topic.

Reviewer 2 Report

Very interested and good written article.

Author Response

Thank you for reviewing the article and for your kind comment.

Reviewer 3 Report

Ortega et al., in their review article Epigenetic alterations related to gestational diabetes mellitus review how the process of Gestational diabetes mellitus (GDM) a metabolic complication in pregnancy, affects the future health of both the mother and the newborn.

The review is well written and easy to follow. The data review is compelling. It will be interesting if the authors can co-relate or discuss HB1Ac levels and methylation or any epigenetic alterations.

The authors focus on epigenetics modifications. Is there any data looking at the proteins which make these epigenetic marks such as DNMTs, Histone acetyltransferase, or histone demethylases that regulate this mark? Is there any correlation with GDM? The addition of this area will improve the review article.

Author Response

The authors appreciate the valuable comments of reviewer. In all the articles included in this review, HB1Ac levels were not measured, so it is not possible to discuss their correlation with epigenetic alterations; however, some studies observed correlations between maternal glucose levels with epigenetic alterations (references 18, 20-22, and 24). To our knowledge, there is a study in which the methylation levels of 65 CpG sites (52 associated with genes) of DNA from umbilical cord blood of women with GDM and controls were evaluated, and although differential methylation was observed between the groups, there was no correlation between HB1Ac levels and methylation levels (Haertle L, et al. Epigenetic signatures of gestational diabetes mellitus on cord blood methylation.” Clinical epigenetics vol. 9 28. 27 Mar. 2017, doi:10.1186/s13148-017-0329-3). These data were not included in the review because we did not aim to cover epigenetic alterations in the umbilical cord.

On the other hand, to date, there are no published works regarding the enzymes that regulate epigenetic alterations associated with GDM. This is because the epigenetics of GDM is a relatively new line of research and because studies of this type require more sophisticated methodological designs that involve both in vitro and in vivo experiments. Only the study by Cao, et al. (reference 27 in the review), reported that, in vitro, miR-98 is up-regulated in placentas from women with GDM, and regulates global methylation by increasing the expression of DNA methyltransferase 1. Outside of this study, there is no greater detail about the enzymes responsible for the epigenetic characteristics of GDM.